# Microstructural and Mechanical Characterization of Colloidal Processed WC/(W5Vol%Ni) via Spark Plasma Sintering

**DOI:** 10.3390/ma16134584

**Published:** 2023-06-25

**Authors:** Ahmed-Ameur Zegai, Hossein Besharatloo, Pablo Ortega, Boubekeur Djerdjare, Begoña Ferrari, Antonio Javier Sanchez-Herencia

**Affiliations:** 1Laboratory of Materials Sciences and Engineering (LSGM), Faculty of Mechanical Engineering and Process Engineering, University of Sciences and Technology Houari Boumediene, Bab Ezzouar 16111, Algiers, Algeria; azegai@usthb.dz (A.-A.Z.); bdjerdjare@usthb.dz (B.D.); 2Instituto de Cerámica y Vidrio, CSIC, Calle Kelsen 5, 28049 Madrid, Spain; pablo.ortega@icv.csic.es (P.O.); bferrari@icv.csic.es (B.F.); ajsanchez@icv.csic.es (A.J.S.-H.); 3CIEFMA, Department of Materials Science and Metallurgical Engineering, EEBE, Universitat Politècnica de Catalunya-BarcelonaTech, 08019 Barcelona, Spain

**Keywords:** tungsten carbide, colloidal processing, spark plasma sintering, sintering aid, microstructural characterisation, mechanical characterization

## Abstract

This study investigates the sintering behaviour and properties of WC-based composites in which WC was mixed with W5vol%Ni in concentrations of 10vol% and 20vol%. Colloidal processing in water and spark plasma sintering were employed to disperse the WC particles and facilitate sintering. The addition of W5vol%Ni improved the sintering process, as evident from a lower onset temperature of shrinkage determined through dilatometric studies. All samples exhibited the formation of tungsten monocarbide (W_2_C), with a more pronounced presence in the WC/20(W5vol%Ni) composite. Sintering reached its maximum rate at 1550 °C and was completed at 1600 °C, resulting in a final density exceeding 99.8%. X-ray diffraction analysis confirmed the detection of WC and W_2_C phases after sintering. The observed WC content was higher than expected, which may be attributed to carbon diffusion during the process. Macro-scale mechanical characterisations revealed that the WC/10(W5vol%Ni) composite exhibited a hardness of 18.9 GPa, while the WC/20(W5vol%Ni) composite demonstrated a hardness of 18.3 GPa. Increasing the W5vol%Ni binder content caused a decrease in mechanical properties due to the formation of W_2_C phases. This study provides valuable insights into the sintering behavior and properties of WC/W5vol%Ni composites, offering potential applications in extreme environments.

## 1. Introduction

The properties of tungsten and its compounds, including carbides, borides, and nitrides, make them widely sought-after materials in various industries such as machining tools and electronics. These materials possess unique characteristics such as high density, high melting point, hardness, chemical resistance, thermal stability, low coefficient of friction, and good electrical conductivity [1,2,3]. Given these qualities, tungsten-based materials are being considered for nuclear applications such as nuclear fusion due to their high density and thermo-mechanical stability, which allows them to withstand the extreme conditions of neutron damage and temperature caused by the plasma. Tungsten and its alloys have been chosen as the material for critical systems, such as divertors, in nuclear fusion reactors such as ITER and DEMO [4,5,6]. Nonetheless, the current tungsten grades have a significant drawback, which is their relatively high ductile-to-brittle transition temperature. To overcome this issue, it is desirable to operate these materials at high temperatures, which can help maintain the integrity of tungsten components while increasing the efficiency of the reactor [7]. However, in the event of an accidental loss of coolant, temperatures can rise to 1200 °C, causing rapid oxidation of tungsten and the release of volatile, radioactively activated tungsten oxides [8]. To enhance the performance of metallic tungsten as a facing wall in the divertor, the use of tungsten carbides and borides has been suggested [9]. These materials offer similar advantageous properties to tungsten, such as high density and melting point, along with additional profits arising from their ceramic nature, such as high hardness [1,10], great flexural strength at high temperatures [11], and slower oxidation kinetics than tungsten [8,12]. WC has shown promise in improving the shielding efficiency of spherical tokamak fusion reactors thanks to its low neutron and gamma attenuation lengths [13,14,15]. However, tungsten carbides and borides pose difficulties in sintering and have low levels of ductility and toughness, which ultimately restrict their range of applications. Therefore, to address mentioned issues, WC-based composites with metallic binders have been considered as a solution to enhance sinterability and densification, while also achieving a remarkable balance between hardness and fracture toughness [16,17] which make them even well-suited for neutron shielding [13,18].

Cobalt has been widely used as a binder for tungsten carbide due to its superior wettability and strong interphase. However, concerns regarding health risks (toxicity), economic fluctuations in pricing and strategic challenges (as cobalt is considered a critical raw material) have made it increasingly difficult for both academic and industrial sectors to substitute traditional WC-Co hard metals. Moreover, cobalt is not permitted for use in civil nuclear applications due to its capacity to absorb neutrons and become highly radioactive. Additionally, the temperature limit for cobalt usage is 800 °C [19,20,21,22,23]. Nickel has emerged as a promising alternative to cobalt, as it offers good wettability [24], lower toxicity, and better price compared to cobalt, along with improved corrosion resistance [25]. Moreover, even small amounts of nickel have been proven to be highly effective in facilitating the sintering temperature of tungsten carbide powders [26].

The mechanical properties of WC-based cermets, including hardness and thermomechanical fatigue at high temperatures, are primarily determined by the carbide phase [27]. Hence, various methods have been suggested to produce a nearly fully dense WC-based cermet suitable for extreme environments. For instance, Spark Plasma Sintering (SPS) has been proposed as a technique for producing WC/W composites, which are composed of two refractory materials and the sinterability of WC/W composite is softer than the binderless WC [11,28]. These composites employ a reactive sintering mechanism, resulting in the production of W_2_C/WC materials that exhibit superior mechanical properties even at high temperatures of up to 1200 °C [26]. While the incorporation of W into WC does enhance densification, it does not result in any significant softening of the sintering conditions at similar temperatures.

As previously mentioned, the addition of small quantities of nickel powder —which are well dispersed within the WC—acts as a sintering aid by reducing the sintering temperature. This allows for the retention of both the material density and mechanical properties while reducing the required metallic binder content [26]. Additionally, small amounts of nickel (less than 1 wt%) are used as sintering activators for metallic tungsten, significantly decreasing the required sintering conditions [29].

The aim of this study is to produce a highly dense WC-based cermet using a pure metallic tungsten binder with a low amount of nickel (W5vol%Ni) to aid in the sintering process. The addition of nickel is expected to initially reduce the sintering temperature of pure tungsten by alloying with the metal, and then reacting with WC to form the pure ceramic WC/W_2_C compound. To achieve this, the study utilises colloidal processing and SPS to improve the distribution of phases and enhance sinterability.

## 2. Materials and Methods

### 2.1. Starting Powders and Mixture Compositions

The study employed commercially available powders as starting materials, including tungsten carbide (Hyperion Materials and Technologies) with a mean particle size of 4.8 µm, metallic tungsten with a primary particle size of 0.2 µm (H.C Starck, Goslar, Germany), and Ni particles with 0.2 µm size aggregated in filaments of about 1.7 µm (INCO 210H, Mississauga, Ontario, Canada). A mixture containing 5 vol% Ni in tungsten was prepared by colloidal processing in water, following a procedure described elsewhere [30]. Figure 1 shows scanning electron microscopy (SEM) images of the initial powders.

Both WC and W5Ni powder mixtures were prepared using high solid content suspensions (30 vol%) using deionised water as a dispersion medium and polyethyleneimine (PEI Mw—2000, Sigma-Aldrich Química S.A., Madrid, Spain) as dispersant [26,28]. The powder mixtures of WC/10(W5Ni) and WC/20(W5Ni) were obtained by mixing the W5Ni and WC powders in the correct proportions as listed in Table 1.

To prepare the mixtures, the W5Ni powders (minor phase) were first added to deionised water containing the calculated amount of PEI, under mechanical stirring. It has to be noted that the metallic powder mixture already contained a certain amount of the dispersant (PEI) from the mixture preparation. Once the mixture of metals was redispersed in water, the WC (main phase) was added. Subsequently, the suspension was ball-milled for 1 h in a plastic jar with silicon nitride balls [31]. The resulting slurries were then dried under a vacuum using a rotary evaporator (Heidolph GmbH, Schwabach, Germany) at approximately 50 °C for 1 h. Finally, the dried powders were crushed in an agate mortar and sieved.

### 2.2. Spark Plasma Sintering (SPS)

The spark plasma sintering (SPS) method was employed to sinter the powders using a Dr. Sinter, SPS-510 CE (Fuji Electronic Industrial Co., Ltd., Shizuoka, Japan) with a 30 mm diameter die and graphite punches. The thermal cycles were adjusted to 1800 °C for the WC-based cermet and 1900 °C for binderless WC with a pressure of 60 MPa. The heating and cooling rates were maintained at 100 °C/min, and a dwell time of 10 min at the maximum temperature was ensured. Figure 2a depicts a schematic illustration of the SPS process, while Figure 2b displays a thermal cycle and pressure sketch for the WC-W5Ni sample sintered at 1800 °C.

The displacement of punches was recorded during the sintering experiments, and the temperature was monitored using an optical pyrometer, which began detecting the sample temperature at 570.3 °C. The dimensional change of the sample during sintering as a function of temperature was obtained from displacement data and was corrected for the expansion of the punches and equipment using a test carried out under the same conditions for a dense WC disk.

After sintering, the samples were taken out of the die and any graphite foil on their surface was removed by brushing. The density of the sintered samples was then determined using the Archimedes method in distilled water following the ASTM B962-08 standard [32]. The theoretical density of different composites was calculated using the rule of mixture, taking into account the detected volume fraction of different phases through the X-ray diffraction (XRD) analysis and the crystallographic density derived from the PDF (Powder Diffraction File) cards of WC, W_2_C, and Ni. The volume fraction of each phase was multiplied by its respective density, and the values were summed to determine the theoretical density.

### 2.3. Microstructural and Mechanical Characterisation of the Sintered Samples

The X-ray diffraction (XRD) patterns were acquired for the powders and sintered samples utilising a Siemens-Bruker D8 Advance Diffractometer (Mannheim, Germany) equipped with a Cu source and a silicon monochromator. The measurement was conducted within a 2θ range of 10° to 90° with a step size of 0.02° (in 2θ). The obtained X-ray diffraction data were analysed using the DIFFRAC.EVA software (Bruker AXS, Billerica, Massachusetts, USA). Semiquantitative analysis was performed by the reference intensity ratio method using the PDF cards 25-1047, 35-0076, and 04-0806 for WC, W_2_C, and W, respectively. Starting powders and sintered samples were analysed using a field emission scanning electron microscope (FE-SEM, Hitachi S-4700, Tokyo, Japan).

After the removal of graphite, Circular-shaped specimens were tested using the Impulse Excitation Technique (IET) at room temperature, in accordance with the ASTM E1876-01 standard [33], utilising Grindo Sonic (MK 5, Belgium). The young and shear modulus were determined by conducting flexural and torsional resonant vibrations, respectively.

Subsequently, the sintered samples were cut with a diamond disc, and the cross-sectional area was polished with diamond pastes, down to 1 μm. The polished surfaces were subjected to X-ray diffraction analysis and FE-SEM microscopy.

Vickers hardness (HV) was measured using a diamond Vickers indenter and applied loads 10 kgf (15 s hold time) by means of a Hardness Tester (LECO V-100 A, St. Joseph, MI, USA). Consequently, indentation fracture toughness (*K_IC_*) was assessed using Anstis et al.’s formula (Equation (1)), based on the length of cracks emanating out of the corner of the Vickers imprints [34].
(1)$$K_{IC}=0.016{\left(\frac{E}{H}\right)}^{0.5}\left(\frac{P}{C^{1.5}}\right)$$
where *E* represents the elastic modulus measured in GPa, *H* denotes the hardness in GPa, *P* signifies the indentation load in N, and *C* indicates the crack length in meter. Residual imprint dimensions and cracks’ length were measured using the optical Microscope Axiophot (Zeiss, Germany).

## 3. Results and Discussions

### 3.1. Powders Characterisation

The FE-SEM micrographs of WC/10(W5Ni) and WC/20(W5Ni) powders after ball milling, drying, and sieving are presented in Figure 3. The SEM images indicate a homogeneous dispersion of both WC (represented by the red line) and W5Ni (represented by the green line) powders. Additionally, it is apparent that the content of the W5Ni metallic phase in WC/20(W5Ni) (Figure 3c,d) is greater compared to that in WC/10(W5Ni) (Figure 3a,b). At higher magnifications (Figure 3b,d), WC and W5Ni can be easily distinguished, indicating that they are attached and form a homogeneous powder composite. Well-distribution of the phases may guarantee that a uniform and homogeneous microstructure can be obtained after sintering.

The XRD pattern of the milled, dried, and sieved powders is presented in Figure 4, indicating the absence of any oxide phases. Additionally, the powders obtained through the described dispersion and mixing process exhibit no signs of corrosion. However, due to the lower atomic mass of nickel compared to W and WC and its relatively low quantity, the Ni peak was not detected in the XRD analysis. 

The intense peaks of WC and W (2θ = 32°, 2θ = 36°, 2θ = 40°, and 2θ = 49°) in sample WC/20(W5Ni) (Figure 4b) exhibit a slight shoulder, which is likely due to the distortion of the crystallographic lattice during the ball milling process.

### 3.2. Spark Plasma Sintering of the Composite Powders

The composite powders and pure WC were sintered by spark plasma sintering and the dimensional change was recorded during the process. The linear shrinkage curve for the binderless WC sample sintered up to 1900 °C and the two mixtures up to 1800 °C is shown in Figure 5a, while Figure 5b presents the corresponding derivative curves as a function of temperature. For comparative purposes, the data reported by Garcia-Ayala et al. for WC/10W and WC/20W compositions were included [28]. The densification process in pure WC starts at 1450 °C, as shown by a change in the slope of the derivative curve. The maximum shrinkage rate—represented by a minimum in the derivative curve—occurs at 1750 °C. At 1850 °C the shrinkage rate slows down drastically, indicating the end of the densification process. However, a small shrinkage is still observed, which is attributed to the removal of the remaining closed porosity. The final density of the sample, measured as 15.4 gr/cm^3^, and corresponds to 98.5% of the theoretical density.

In addition, Figure 5a displays the dilatometric curves for the sintering of the composites containing 10% and 20% W5Ni powder mixtures, represented by the red and blue curves, respectively. Comparing the two compositions with and without nickel, a significant decrease in the characteristic temperatures is evident during sintering. In the case of WC20(W5Ni), the sintering process appears to begin at 1000 °C, while for WC10(W5Ni), it occurs at 1100 °C. However, the derivative curves in Figure 5b indicate that the start of shrinkage occurs at a lower temperature, between 850 °C and 900 °C. The initial shrinkage observed in the composite containing nickel is attributed to the deformation of the metallic phase and the activated sintering of the W-Ni mixture [35] and does not involve the participation of the WC phase. Furthermore, it is apparent that the inclusion of Ni accelerates the sintering process in the 80WC/20W and 90WC/10W compositions, leading to a faster decrease in shrinkage compared to compositions without Ni. Both WC/10(W5Ni) and WC/20(W5Ni) samples exhibited changes associated with spark plasma reactive sintering (SPRS) around 1200 °C that led to the formation of the semi-carbide (W_2_C) (as shown in Figure 5b). Following this reaction, the sintering continued, and the maximum shrinkage rate occurred at 1540 °C for both samples. Subsequently, the shrinkage process finished at 1650 °C for both compositions, resulting in a final contraction of 42% and 47% for WC/10(W5Ni) and WC/20(W5Ni), respectively. These values are higher than the 37% observed for the binderless WC sample. The final densities achieved were 15.86 and 16.22 for WC/10(W5Ni) and WC/20(W5Ni), respectively (Table 2).

Figure 6 displays the XRD patterns obtained for WC/10(W5Ni) and WC/20(W5Ni) samples after sintering. In both cases WC and W_2_C phases were identified as unique, implying the complete reaction of W with WC. Table 2 presents the weight percent ratio of each phase, considering both the complete reactions of the reactants and the analysis of the XRD patterns. The density of the samples measured by the Archimedes method and the assessed percentage of the theoretical density is also included, considering the complete reaction of the starting powders and the phases calculated from the crystallographic data. As evident from Table 2, the amount of WC detected by XRD is higher than what was expected from the complete reaction of the W, indicating the diffusion of carbon from the die and punches into the sample.

The WC/10(W5Ni) and WC/20(W5Ni) composites demonstrated nearly full density, as expected from the dilatometric curves, achieving 99.8% and 100% of the theoretical values measured from the XRD analysis, respectively. Interestingly, the density of the WC/W5Ni composites was higher compared to nickel-free samples of WC/W reported in previous works [11,28]. This finding supports the notion that even small amounts of nickel in the mixture (0.3 and 0.5 wt%) play a crucial role in sintering, enabling the production of fully dense materials under softer conditions.

### 3.3. Microstructural Characterisation of Sintered Samples

Figure 7 presents SEM micrographs of the sintered studied samples. The light grey areas represent the W_2_C phase, while the dark grey regions correspond to the WC phase, as confirmed by XRD analysis. In both samples, the presence of pores was not significant, as the high density was achieved.

The 90WC10(W5Ni) sample (Figure 7a–c) exhibits isolated W_2_C zones that are homogeneously distributed throughout the sample, with an average particle size of 5 µm. On the other hand, a different scenario was observed for sample 80WC20(W5Ni) (Figure 7d–f), wherein a higher amount of W_2_C was detected. The dispersion of W_2_C areas exhibited two distinct patterns. Firstly, as mentioned earlier, W_2_C particles with a mean size of approximately 5 µm were homogeneously dispersed throughout the sample. Secondly, larger agglomerations of semi-carbide areas with sizes ranging between 20 and 50 µm were observed. In both compositions, the presence of porosity-like regions was observed among the WC phase after the liquid-phase sintering process. There is speculation that these regions may contain nickel. However, it is important to note that definitive confirmation of these porosity-like regions being composed of nickel cannot be made at this stage. Further comprehensive studies are required to validate this assumption and provide conclusive evidence. However, such regions were not observed among W_2_C, indicating a solid-state sintering process.

Considering above mentioned data, a sintering mechanism can be proposed for the studied composites. In the first stage at 900 °C, the presence of nickel activates the sintering process and melting of the W/Ni alloy. At 1200 °C, the melted W reacts with the WC phase diffuses resulting in the formation of solid W_2_C that continues densification by a solid-reactive sintering mechanism while leaching the nickel out of the composition. Subsequently, the melted nickel induces the liquid phase sintering process for the WC. The WC dissolves in the melt and precipitates in the faceted microstructure, while the nickel is left in the triple points. Finally, it is anticipated that all the added nickel from the W(5Ni) powder mixture only appears with the WC phase, which has been partially consumed by the W to produce the W_2_C phase. As a result, the ratio of Ni/WC increases to 0.6 vol% in the 90WC10(W5Ni) sample and 1.8 vol% in the WC/20(W5Ni) one. These differences in nickel content are also evident in the micrographs at high magnifications (Figure 7c,f).

### 3.4. Mechanical Characterisation of Sintered Samples

The measured mechanical properties of the studied samples, the reported WC, and W_2_C are summarised in Table 3. In general, the mechanical properties —including hardness, elastic modulus, and shear modulus—of WC/20(W5Ni) were observed to decrease compared to WC/10(W5Ni) based on the data presented in Table 3. This trend can be attributed to the higher quantity of W_2_C phase formed during the sintering process, as reported in Table 2. The decrease in mechanical properties based on the quantity of the W_2_C phase can be explained by various factors. Firstly, WC has a primarily covalent bonding nature, while W_2_C has a partially covalent and partially metallic bonding nature [36,37,38]. As a result, the W_2_C phase is softer compared to WC, leading to an overall decrease in composite hardness. Secondly, the formation of W_2_C can lower the volume fraction of the load-bearing WC phases, causing a decrease in the elastic modulus [39,40,41,42,43]. Furthermore, the HV and elastic modulus values of the studied samples fall between the reported values for WC and W_2_C, which suggests that the mechanical properties of the material are influenced by the composite-like behaviour of the studied composition following the rule of mixture. The rule of mixture is a predictive model that estimates the overall mechanical properties of a composite material by considering the properties of its constituent materials and their respective volume fractions (determined through XRD Analysis). To gain more accurate and detailed information about the mechanical properties of the studied samples, further investigation employing techniques such as nanoindentation mapping is necessary [44,45].

The fracture toughness values of the sintered composites are higher compared to the values of binderless WC and W_2_C single phase reported in the literature, this can be explained by the presence of the small content of Ni dispersed homogeneously into the microstructure, where Ni is directly proportional to fracture toughness [42]. 

## 4. Conclusions

WC/W_2_C composites were prepared by mixing WC with 10 and 20 Vol% of W5Ni using a colloidal process. The mixture of powders was then sintered using spark plasma sintering, followed by microstructural and mechanical characterisation. In conclusion, this study reports successful colloidal processing to achieve a homogeneous distribution of powder mixtures without the formation of oxide phases. After spark plasma sintering, all metallic W reacted with WC to form W_2_C in each composite, as confirmed by dilatometric studies and X-ray diffraction analysis. Samples containing nickel showed lower temperature shrinkage and facilitated the sintering process, which resulted in a true density of over 99.9% and good phase dispersion as seen in microstructure imaging. These findings suggest that the addition of nickel to the metallic W binder could improve the sintering process and the resulting properties of WC-based composites. 

In terms of mechanical properties, the hardness and young modulus of composites exhibit an inverse relationship with the binder content. An increase in W5Ni binder content results in a reduction in mentioned mechanical properties because it leads to the formation of W_2_C phases that are less stiff and less hard than WC. Furthermore, despite the high concentration of WC and W_2_C phases and the brittle nature of these phases in the sintered samples, the fracture toughness results of samples containing a low Ni content surpass those of WC and W_2_C single phases reported in the literature.

## Figures and Tables

**Figure 1 materials-16-04584-f001:**
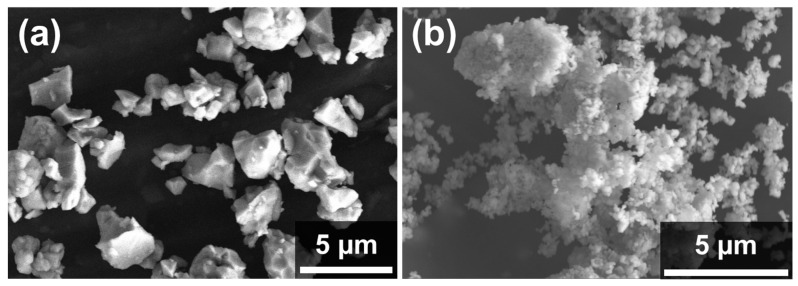
SEM micrograph of starting powders (**a**) WC, (**b**) W5Ni.

**Figure 2 materials-16-04584-f002:**
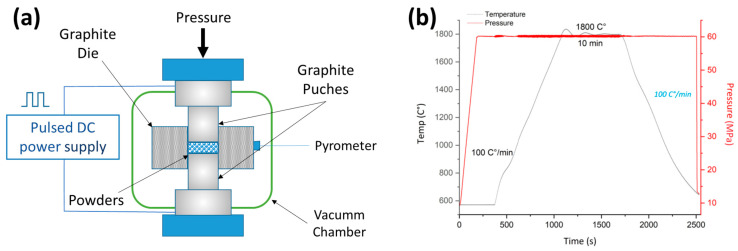
(**a**) Scheme of SPS machine. (**b**) Thermal cycle of WC-W5Ni mixtures.

**Figure 3 materials-16-04584-f003:**
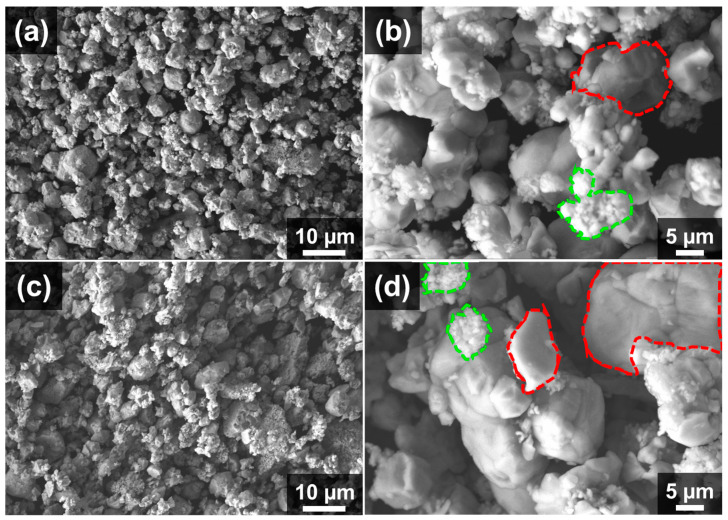
FESEM micrographs of composite powders at different magnifications: (**a**,**b**) WC/10(W5Ni). (**c**,**d**) WC/20(W5Ni). The red-dashed lines indicate WC particles, while the green-dashed regions represent W5Ni.

**Figure 4 materials-16-04584-f004:**
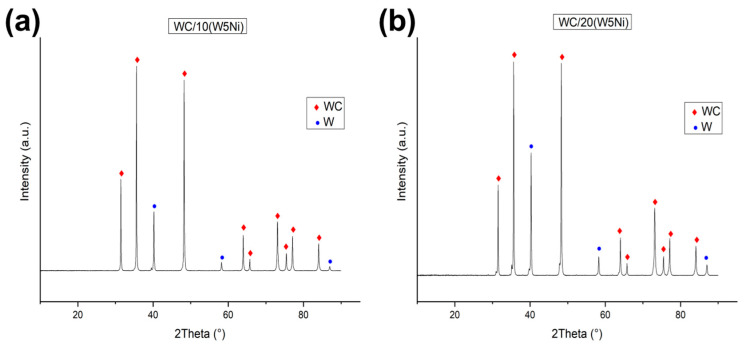
XRD patterns of powders mixtures, (**a**) WC/10W5Ni powder, (**b**) WC/20W5Ni.

**Figure 5 materials-16-04584-f005:**
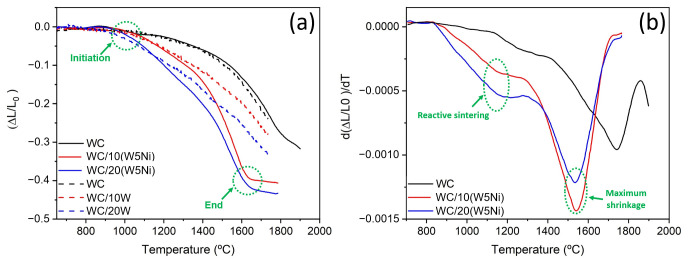
(**a**) Linear shrinkage vs. temperature plots recorded during the SPS process, (**b**) The differential curves.

**Figure 6 materials-16-04584-f006:**
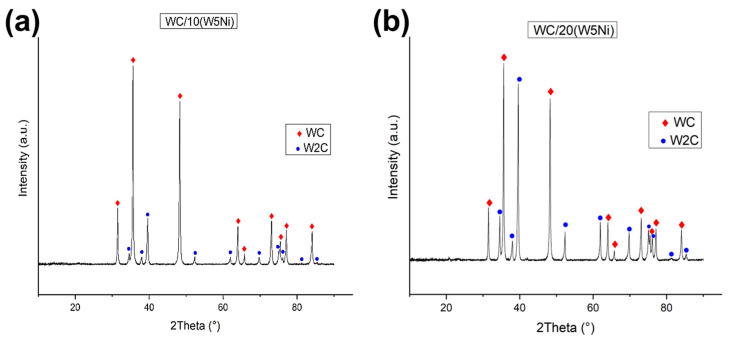
XRD analysis of sintered samples: (**a**) WC/10(W5Ni) and (**b**) WC20(W5Ni).

**Figure 7 materials-16-04584-f007:**
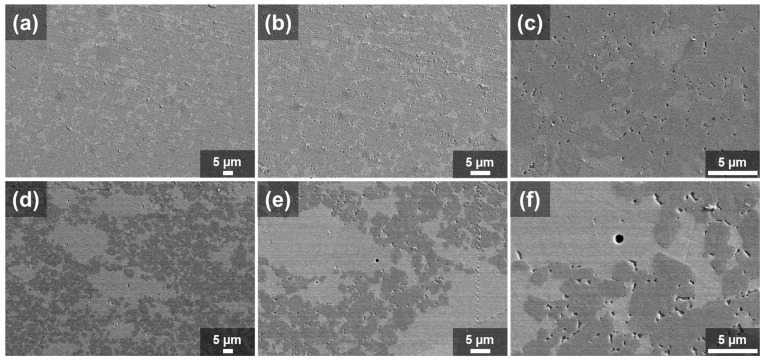
SEM micrograph sintered samples at different magnifications (**a**–**c**) WC/10W5Ni. (**d**–**f**) WC/20W5Ni.

**Table 1 materials-16-04584-t001:** Designation and composition of studied WC-based cermet.

Compositions	WC + Binder (vol%)			WC + Binder (wt%)		
	WC	W	Ni	WC	W	Ni
WC/10(W5Ni)	90	9.5	0.5	88.25	11.47	0.28
WC/20(W5Ni)	80	19	1	76.94	22.51	0.55

**Table 2 materials-16-04584-t002:** Archimedes density, and true density of sintered compositions with the relative bulk and true density.

Composition	True Density	Complete Reaction			XRD Analysis		
		WC	W_2_C	Density (g/cm^3^)	WC	W_2_C	Density
WC/10(W5Ni)	15.86	76.2%	23.8%	16.02 (99.0 th%)	86.4%	13.6%	15.89 (99.8 th%)
WC/20(W5Ni)	16.22	53.3%	46.7%	16.30 (99.5 th%)	62.1%	37.9%	16.20 (99.9 th%)

**Table 3 materials-16-04584-t003:** Mechanical properties of the samples reported in this study and literature [39,40,41,46].

Composition	H_V_ (GPa)	K_IC_ (MPa.m^1/2^)	E (GPa)
WC/10(W5Ni)	18.9 ± 0.3	9.3 ± 0.7	663 ± 15
WC/20(W5Ni)	18.3 ± 0.3	9.5 ± 1.0	575 ± 15
WC	24–28	6	706
W_2_C	17.1	3.6	420–444

## Data Availability

The datasets generated during the current study are available from the corresponding author on reasonable request.

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
