# Peer review of "Microstructural and Mechanical Characterization of Colloidal Processed WC/(W5Vol%Ni) via Spark Plasma Sintering"

_materials, 2023, doi:10.3390/ma16134584_

Round 1

Reviewer 1 Report

The article belongs to the field of fusion materials science and will be of interest to those who are trying to make various tungsten based composites that are better suited for plasma facing components. I have only a few comments and suggestions.

Lines 124-131 The paragraph about XRD should be moved to section 2.3, with removing "mechanical" from the title

Line 164 it is desirable to add the formula itself to the text

Line 170-172 what method was used to identify the elemental (phase?) composition of particles

Line 240 you need to specify links (even if it will be a repetition of the introduction)

Line 244 (table 2) Density units are not specified

Lines 247-249 What method was used to identify the phase composition in SEM images (type of SEM detector, etc.) and what exactly did XRD confirm?

Reviewer 2 Report

ِDear editor in chief

I owe you an aplogize for sending late reply .

this paper have a good results and can be publish in this form.

Best wishes

ِDear editor in chief

I owe you an aplogize for sending late reply .

this paper have a good results and can be publish in this form.

Best wishes

Reviewer 3 Report

1.       Line 126 "Cu Kα radiation (λ=1.540598 Å)". Does it make sense to specify so precisely the wavelength of the doublet? Also the absence of diffraction maxima from Kα2-radiation on the diffraction pattern is not clear.

2.       The SEM images (fig. 3) show WC and W5Ni. How can these phases differ in the images.

3.       What are the additional maxima to the left of the diffraction peaks (fig.4)?

4.       Line 187 - "crystallographic net" is not conventional term

5.       Line 233. Is there any other evidence of possible diffusion of carbon into the sample?

6.       Please add the experimental densities and theoretical density calculation algorithm.

7.       In the introduction it was shown that it was appropriate to use nickel instead of cobalt, it would be worthwhile to compare the sintering kinetics and properties of samples made with cobalt in the discussion.

Please check the terms used.

Reviewer 4 Report

Thanks to the esteemed authors for the submission of the paper to the journal Materials!

The paper "Microstructural and Mechanical Characterization of Colloidal Processed WC/(W5Vol%Ni) via Spark Plasma Sintering" is devoted to sintering material at lower temperature by mixing WC with W5Vol%Ni in concentrations of 10vol% and 20vol%. Phase separation and sintering were achieved by colloidal treatment in water and SPS, respectively. The work has merit for presentation to the scientific community and can be accepted for publication after some improvements.

Comments on the paper are presented below for each section of the manuscript.

Abstract

(1) The first sentence of the abstract should give a brief and concise summary of the work carried out. For the reader's convenience the abstract should be accompanied by the exact numerical values of the data obtained from the mechanical tests performed.

Introduction

(2) The literature of the last three years can be used to expand and supplement the topicality, theoretical background and rationale for the purpose of the study. This also includes the addition of sources published in the Materials journal to the list of references.

(3) The first paragraph of the introduction is structured without references. Obviously, these should be reviews of functional WC-based alloys.

(4) The emphasis placed by the authors on the possible application of their WC alloying approach to realizing nuclear applications should be considered as the most important of the list of applications. The introduction section could be greatly improved and the high scientific value of the paper further substantiated if the authors could expand on this topic by presenting and analyzing the achievements and limitations of the actual known cases of WC alloy applications in this field.

Materials and Methods

(5) The methodological part presents the methodological details of the approach for the preparation of starter composite mixtures, which is borrowed from [25]. The authors used denoised water as a medium for preparing the starting powders, which can have a significant effect on the corrosion status of the metal binder in the composition. It is stated in lines 181-182 that "The XRD pattern of the ground, dried and sieved powders is shown in Figure 4, 181 which indicates the absence of any oxide phases". It would be advisable for the authors to also state in their paper that the powders obtained do not show any signs of corrosion under the given dispersion and mixing conditions. Perhaps this is a justification for the fact that anhydrous solutions are not necessary for this method.

(6) There is a lack of data in the paper on the measurement of particle size distribution in powders.

(7) A schematic diagram of the thermomechanical sintering process by SPS is shown in Fig. 2a. An installation of the Dr Sinter, SPS-510 CE type from Fuji is used. From the diagram (Fig. 2a) it can be seen that the pressure is applied from both the top and the bottom. It is most likely that the moving part of the unit is only the lower part.

(8) Lines 148-149 state that "...graphite foil on their surface was removed by brushing...". The complete removal of the graphite from the surfaces of the samples obtained via SPS at the temperature mode higher that 1500°C by means of brushing alone is questionable. This is due to the deep diffusion of carbon into the material. The carbon in such a case can only be removed by grinding a minimum of 100–200 µm of the surface.

(9) It would be interesting to obtain information on the dispersion of the microhardness values of the surface of the samples. This could indirectly indicate the structural heterogeneity of the material using the criterion of local hardness in the homogeneity of the phase distribution.

(10) The results of testing for other mechanical properties, such as the ISO flexural strength of the best specimen, are not presented in the paper.

(11) The average grain size is not known. It is therefore not possible to estimate the contribution of temperature to the agglomeration of the sintered particles.

(12) A semi-quantitative calculation of the number of major phases using the Rietveld method can be added to the XRD results.

Results and discussion

(13) Figures 3b and 3d need to be supported by EDS elemental mapping. Also, the addition of the EDS results to Figure 7 is recommended. Synthesis parameters of 1800 and 1900 °C have been chosen for sintering. These are significantly higher than the sintering temperatures of WC and Ni known from the literature (10.1016/j.jallcom.2019.152547, 10.1016/j.ijrmhm.2021.105725). The melting point of nickel is found to be 1455 °C. At these high sintering temperatures, the binder must be overheated and consequently have leakage outside the perimeter of the sample. This should occur at an early stage of sintering, taking into account the model according to which the mobile distribution of the binder (Co, Fe, Ni) occurs at lower temperatures under SPS conditions compared to conventional consolidation methods (please check: doi:10.1016/j.msea.2008.01.048. Line 273 - states that "These differences in nickel content are also evident in the micrographs at high magnification". However, the paper does not have any SEM images in support of this statement. Please comment on this issue in the text and compare the structural organization of the alloys obtained with their counterparts obtained by SPS at lower sintering temperatures in the range of 1150-1200 °С.

(14) An analysis of the criterion of the obtained properties of the synthesis material is missing from the work. This can be presented in the form of a comparative table, where alloys of the presented group, obtained by different methods from raw materials of different composition and size range of starting components, are compared in terms of average grain size, achieved relative density (expressed as % of theoretical), hardness, flexural strength, fracture toughness, Young's modulus.

(15) In Figure 5, it would be useful to have an indication of the sintering stages.

Reviewer 5 Report

This study addressed the issue by mixing WC with W5vol%Ni in 10vol% and 20vol% concentrations. Colloidal processing in water and spark plasma sintering were employed for phase dispersion and sintering, respectively. The addition of nickel improved the sintering process, evident from a lower onset temperature of shrinkage determined through dilatometric studies. There are still some problems that need further improvement.

1. Why is volume concentration chosen instead of mass concentration in the design of W5%volNi? As well as solid content suspension (30vol%) instead of g/L?

2. Why did author choose solution mixing? Is there metal oxidation after the solution mixing? Such as NiO.

3. The authors should give the design components and actual components, and suggest adding EDS or EPMA characterization results.

4. How are the phases of different regions determined in Figure 7? EDS point analysis is recommended.

5. What form of Ni exists in the sintered samples? Solution, precipitation, or inclusion?

6. It is suggested to add EBSD to study the influence of flux on the grain size of sintered samples.

Round 2

Reviewer 3 Report

Dear authors Authors and Editors,

The paper was revised and clarified with my comments. Could be accepted in present form.

Reviewer 4 Report

Thanks to the authors for the work done! All comments have been answered comprehensively. The manuscript has been modified, increasing the quality of the paper.

The paper can be accepted in its current form.

Reviewer 5 Report

The author did not provide scientific evidence for my question.

1. For many experiments, the designed component and the actual component are not exactly the same, and the actual component need to be tested to determine.

2. W2C and WC have different W/C atomic ratios, EDS can do quantitative or semi-quantitative analysis, the author can not rely on subjective consciousness and experience to determine the phase.

3. There is no experimental evidence for the existence form of Ni.
